# Intramyocardial Inflammation after COVID-19 Vaccination: An Endomyocardial Biopsy-Proven Case Series

**DOI:** 10.3390/ijms23136940

**Published:** 2022-06-22

**Authors:** Christian Baumeier, Ganna Aleshcheva, Dominik Harms, Ulrich Gross, Christian Hamm, Birgit Assmus, Ralf Westenfeld, Malte Kelm, Spyros Rammos, Philip Wenzel, Thomas Münzel, Albrecht Elsässer, Mudather Gailani, Christian Perings, Alae Bourakkadi, Markus Flesch, Tibor Kempf, Johann Bauersachs, Felicitas Escher, Heinz-Peter Schultheiss

**Affiliations:** 1Institute of Cardiac Diagnostics and Therapy, IKDT GmbH, 12203 Berlin, Germany; ganna.aleshcheva@ikdt.de (G.A.); dominik.harms@ikdt.de (D.H.); ugross@zedat.fu-berlin.de (U.G.); felicitas.escher@charite.de (F.E.); heinz-peter.schultheiss@ikdt.de (H.-P.S.); 2Kerckhoff Heart Center, Department of Cardiology, 61231 Bad Nauheim, Germany; kardiologie@kerckhoff-klinik.de; 3Department of Cardiology and Angiology, Universitätsklinikum Gießen und Marburg, 35391 Gießen, Germany; birgit.assmus@innere.med.uni-giessen.de; 4Department of Cardiology, Pulmonology and Vascular Medicine, Heinrich Heine University Düsseldorf, 40225 Düsseldorf, Germany; ralf.westenfeld@med.uni-duesseldorf.de (R.W.); malte.kelm@med.uni-duesseldorf.de (M.K.); 5Onassis Cardiac Surgery Center, 176 74 Athens, Greece; srammos@gmail.com; 6Department of Cardiology, University Medical Center Mainz, 55131 Mainz, Germany; wenzelp@uni-mainz.de (P.W.); tmuenzel@uni-mainz.de (T.M.); 7Department of Cardiology, Klinikum Oldenburg, 26133 Oldenburg, Germany; elsaesser.albrecht@klinikum-oldenburg.de; 8Frankenwaldklinik, 96317 Kronach, Germany; mudather.gailani@helios-gesundheit.de; 9Department of Cardiology, St. Marien-Hospital, 44534 Lünen, Germany; m1@klinikum-luenen.de; 10Department of Internal Medicine, Cardiology, Geriatrics and Palliative Medicine, Gemeinschaftsklinikum Mittelrhein gGmbH, 56727 Mayen, Germany; alae.bourakkadi@gk.de; 11Department of Cardiology, Marienkrankenhaus gGmbH, 59494 Soest, Germany; m.flesch@mkh-soest.de; 12Department of Cardiology and Angiology, Hannover Medical School, 30625 Hannover, Germany; kempf.tibor@mh-hannover.de (T.K.); ratic.angelina@mh-hannover.de (J.B.); 13Department of Cardiology, Campus Virchow-Klinikum, Charité University Medicine Berlin, 13353 Berlin, Germany; 14German Centre for Cardiovascular Research (DZHK), Partner Site Berlin, 10785 Berlin, Germany

**Keywords:** COVID-19, vaccination, SARS-CoV-2, Comirnaty, Vaxzevria, Janssen, inflammatory cardiomyopathy, myocarditis, giant cell myocarditis

## Abstract

Myocarditis in response to COVID-19 vaccination has been reported since early 2021. In particular, young male individuals have been identified to exhibit an increased risk of myocardial inflammation following the administration of mRNA-based vaccines. Even though the first epidemiological analyses and numerous case reports investigated potential relationships, endomyocardial biopsy (EMB)-proven cases are limited. Here, we present a comprehensive histopathological analysis of EMBs from 15 patients with reduced ejection fraction (LVEF = 30 (14–39)%) and the clinical suspicion of myocarditis following vaccination with Comirnaty^®^ (Pfizer-BioNTech) (*n* = 11), Vaxzevria^®^ (AstraZenica) (*n* = 2) and Janssen^®^ (Johnson & Johnson) (*n* = 2). Immunohistochemical EMB analyses reveal myocardial inflammation in 14 of 15 patients, with the histopathological diagnosis of active myocarditis according the Dallas criteria (*n* = 2), severe giant cell myocarditis (*n* = 2) and inflammatory cardiomyopathy (*n* = 10). Importantly, infectious causes have been excluded in all patients. The SARS-CoV-2 spike protein has been detected sparsely on cardiomyocytes of nine patients, and differential analysis of inflammatory markers such as CD4^+^ and CD8^+^ T cells suggests that the inflammatory response triggered by the vaccine may be of autoimmunological origin. Although a definitive causal relationship between COVID-19 vaccination and the occurrence of myocardial inflammation cannot be demonstrated in this study, data suggest a temporal connection. The expression of SARS-CoV-2 spike protein within the heart and the dominance of CD4^+^ lymphocytic infiltrates indicate an autoimmunological response to the vaccination.

## 1. Introduction

Myocarditis and pericarditis have been reported as a potential complication of the COVID-19 mRNA (messenger RNA) vaccines. Since the first observations of myocarditis in response to the Pfizer-BioNTech vaccine in April 2021 [1], numerous case reports, indicating a relationship between COVID-19 vaccination and the occurrence of myocarditis/pericarditis, have been published [2,3,4,5,6,7,8].

Meanwhile, large epidemiological studies reveal that the majority of COVID-19-vaccine-related myocarditis occur in young male individuals (median age of 21 years) following the second dose of mRNA based vaccines [9]. Most cases (76.8%) have been observed in response to the Pfizer-BioNTech vaccine (Comirnaty^®^), whereas one quarter (23.2%) received the Moderna vaccine (Spikevax^®^). Despite total case numbers, general incidence of myocarditis was shown to be higher in patients receiving Spikevax^®^ (1.3 to 1.9 (Spikevax^®^), and 0.26 to 0.57 (Comirnaty^®^) extra cases of myocarditis per 10,000 subjects) [10]. Consequently, numerous national authorities recommended the use of Comirnaty^®^ instead of Spikevax^®^ for people aged 30 years and younger.

Signs of myocarditis after COVID-19 vaccination usually develop within 2 weeks [11,12] and clinical symptoms generally ease quickly without impairment of cardiac function [13]. Although COVID-19-vaccine-related myocarditis has an overall marginal incidence and affected patients show a fast recovery in absence of short-term complications, myocarditis is an alarming side effect, which needs to be monitored carefully.

While the majority of published cases are based on cardiac magnetic resonance imaging (cMRI) and laboratory evaluation [2,3,4,5,6,7], endomyocardial biopsy (EMB)-proven cases are limited [8,14,15]. Here, we present a comprehensive analysis of EMBs from 15 patients with the suspicion of myocarditis after vaccination against SARS-CoV-2 and reveal a temporal relation between the vaccination and the occurrence of myocardial inflammation, ranging from mild inflammatory cardiomyopathy to severe active myocarditis and giant cell myocarditis.

## 2. Results

Patients’ characteristics, clinical picture and laboratory findings at hospital admission are depicted in Table 1. Importantly, the suspicion of myocarditis in response to vaccination against SARS-CoV-2 was expressed in all cases. Median left ventricular ejection fraction (LVEF) was 30 (14–39)%. The analyzed cohort ranges from 18 to 68 years of age and is dominated by male sex (9/15, 60%). All patients were Caucasian. Vaccine type, onset of symptoms, clinical suspicion and clinical course differ a lot in the presented cohort (Table 1).

All EMBs were negatively tested for SARS-CoV-2 using E-gene-specific sequences [16]. Besides 10 cases with latent parvovirus B19 infection, no other viral pathogens including adenovirus, enterovirus, Epstein–Barr virus and human herpesvirus 6 were detected in the biopsies by nested- and qRT-PCR (Table 2). From 15 patients, two cases of active myocarditis (AMC, patients 1 and 2), two cases of giant cell myocarditis (GCMC, patients 3 and 14), nine cases of inflammatory cardiomyopathy (DCMi) and one case of dilated cardiomyopathy (DCM, patient 12) were diagnosed on the basis of EMB differential diagnostics. The majority of cases (11/15, 73%; AMC = 2; GCMC = 1; DCMi = 8) were observed in relation to the Comirnaty^®^ mRNA vaccine, and four cases (27%) were related to Vaxzevria^®^ (DCMi = 1; DCM = 1) and Janssen^®^ (AMC = 1; DCMi = 1) vector vaccines. The onset of symptoms varied from 0 to 56 days (median 14 days) for Comirnaty^®^, from 1 to 14 days (median 7.5 days) for Vaxzevria^®^, and from 14 to 28 days (median 21 days) for Janssen^®^ vaccines.

The majority of patients in the Comirnaty^®^ group were male (8/11, 73%), while suspicion of myocarditis after vaccination with the vector vaccines was mainly made in female patients (1/2 for Janssen^®^; 2/2 for Vaxzevria^®^). The average age of patients receiving the Comirnaty^®^ vaccine was 34 (28–46) years, of patients receiving Vaxzevria^®^ 64 (61–66) years and of patients receiving Janssen^®^ vaccines 38 (35–42) years (Table 1).

All patients revealed a sudden onset of severe left ventricular failure (LVEF ≤ 45%) and most had additional symptoms such as dyspnea, chest pain, reduced respiratory rate and diminished exercise capacity (Table 1). Data on cardiac MRI were available for six patients, showing no signs of myocarditis in four of them (patients 2, 4, 5 and 11) and typical signs of active myocarditis in two patients (patients 8 and 9) (Table 1). Five patients showed severe complications including cardiac decompensation (patients 5 and 8, both DCMi) and fulminant cardiogenic shock (patient 14, GCMC), and two patients (patients 2 and 3, AMC and GCMC, respectively) had to be resuscitated before admission to the intensive care unit (Table 1). Data on laboratory cardiac and inflammatory biomarkers including troponin, brain natriuretic peptide (BNP), creatinine kinase (CK) and C-reactive peptide (CRP) are heterogeneous and were elevated in 12 of 15 patients (Table 1).

Immunohistochemical EMB analysis revealed increased presence of inflammation markers in 14/15 patients (Figure 1, Table 2). Only 1 patient was found to have consistently low levels of inflammatory cells, compatible with the diagnosis of a DCM (patient 12, Figure 1, Table 2). All other patients with myocardial inflammation showed a diverse pattern of inflammatory markers, ranging from mild to severe degree of inflammation. From 14 patients with proof of inflammation, 9 (64%) showed elevated infiltration of CD3^+^ T cells, 12 (86%) showed increased levels of CD45R0^+^ T-memory cells and LFA-1^+^ lymphocytes, 11 (79%) showed increased numbers of MAC-1^+^ macrophages and HLA-DR^+^-presenting cells, and none showed augmented levels of perforin^+^ cytotoxic cells (Figure 1, Table 2). Cellular adhesion molecules ICAM-1 and VCAM-1 were normal in all patients, with the exception of two patient with AMC (patient 2) and GCMC (patient 14), respectively (Table 2). Further immunohistochemical analysis revealed positive detection of the SARS-CoV-2 spike protein in cardiac tissue in a couple of patients. In particular, the spike protein was found in sparse cells (cardiomyocytes) in 9 of 15 cases (Figure 2, Table 2). Moreover, except the cases with active myocarditis (patients 1 and 2), giant cell myocarditis (patients 3 and 14), and one case of DCMi (patient 9), CD4^+^-T-cell-to-CD8^+^-T-cell ratio was ≥1, suggesting a predominantly autoimmunological origin of the observed inflammation (Figure 3, Table 2).

## 3. Discussion

In the present study, we identified 14/15 (93%) patients with suspected myocarditis after COVID-19 vaccination, with EMB-proven myocardial inflammation. Four of them had acute myocarditis (including active myocarditis and giant cell myocarditis), and 10 were diagnosed for inflammatory cardiomyopathy (DCMi). The absence of inflammation in EMBs of one patient might be explained by a sampling error. The onset of symptoms varied from 0 to 56 days, which is within the range of previous epidemiological findings [17]. Even though a causal relationship cannot be made, infectious reasons fro myocarditis had been excluded by molecular diagnostics of the most relevant viral pathogens (including SARS-CoV-2). Although latent viral genomes of B19V were found in 10/15 (67%) cases, active B19V transcription was not confirmed in these patients. B19V is the most frequently detected viral species in the human heart, and its contribution to myocardial inflammation and thus, to long-term patient outcome, is highly dependent on its transcriptional activity [18,19]. Therefore, identified latent B19V infections are unlikely causative of the myocardial inflammation in these patients.

Because viral infections have been ruled out as the cause for myocarditis/myocardial inflammation, autoimmunological mechanisms might be an explanation. Cross-reactivity of spike protein antibodies with myocardial contractile proteins, mRNA immune reactivity and hormonal involvement, have been discussed as potential mechanisms by which COVID-19 mRNA vaccines induce hyperimmunity [20]. In the present cohort, the SARS-CoV-2 spike protein was found to be expressed on cardiomyocytes in 9 of 15 patients. Thus, vaccine-encoded spike protein seems to reach the heart, where it might trigger an inflammatory response, resulting in the development of myocarditis or DCMi. Previous mouse data indicated an intramyocardial expression of spike protein after mRNA vaccination [21]; however, human data are scarce. Interestingly, except for five cases (AMC = 2, GCMC = 2 and DCMi = 1), the number of CD4^+^ T lymphocytes was either equal or higher than those of CD8^+^ T cells. In acute myocarditis, a shift of CD4-to-CD8 ratio towards CD8^+^ T cells is known [22], explaining the elevated CD8^+^ cells in AMC and GCMC patients. However, as CD4^+^ T cells are considered to be the major driver of autoimmune myocarditis [20], our data support the idea that vaccine-induced myocardial inflammation is a consequence of excessive CD4^+^ T-cell infiltration, and thus, a potential driver of autoimmunological myocardial damage. Moreover, the expression of HLA-DR was increased in 11 of 14 (79%) patients with evident inflammation. Due to the fact that HLA class II regions strongly associate with several autoimmune diseases [20], induction of HLA-DR in response to the vaccination supports an autoimmunological contribution to myocardial inflammation after vaccination. Expression of ICAM-1 and VCAM-1 adhesion molecules on the cardiac (micro)vascular endothelium is upregulated upon infection with myocarditis-associated viruses [23]. Besides one patient with AMC (patient 2) and one patient with GCMC (patient 14), none of the patients showed increased levels of ICAM-1 and VCAM-1, supporting a non-infectious origin of the observed inflammation. Perforin-mediated cardiac damage is known to be involved in viral [24,25] and acute idiopathic myocarditis [26]. Interestingly, none of the patients showed increased presence of perforin^+^ cells, indicating no contribution of cytotoxic events following the COVID-19 vaccination. Moreover, data on perforin^+^ cells in patients with reduced LVEF show that the lack of perforin^+^ cardiac infiltrates is associated with spontaneous LVEF improvement [23]. Thus, improvement of cardiac function could be assumed, as described for most of the vaccine-related myocarditis patients [13].

Among EMB diagnostics, non-invasive imaging techniques have been used to identify COVID-19-vaccine-related myocarditis [2,3,4,5,6,7]. Besides cMRI markers such as T2-weighted ratio, early gadolinium enhancement and late gadolinium enhancement within the Lake Louise criteria and somatostatin receptor positron emission tomography/computed tomography (PET/CT) have been applied for the diagnosis of myocarditis following COVID-19 vaccination [27]. However, as cardiac imaging techniques give helpful, but limited information about severe myocardial inflammation, they are unable to correlate with mild inflammation and viral persistence. The present case series reveal that cMRI was able to detect myocardial inflammation in only 33.3% (2/6) of the cases (Table 1). Thus, EMB remains the definite standard in the diagnosis of viral and inflammatory cardiomyopathy.

### Study Limitations

There are several limitations of this study which have to be considered when interpreting the obtained results. First, no causality can be assumed or established due to the lack of a control group and the observational character of a case series. Although other causes of myocardial inflammation have been ruled out, there is no direct evidence of a vaccine-induced inflammatory response in the presented cases. Second, an incidence of vaccine-mediated myocarditis/inflammatory cardiomyopathy cannot be determined due to the selective approach of choosing cases with the suspicion of myocarditis after COVID-19 vaccination and the simultaneous availability of EMBs. Third, there is a lack extended clinical data for all of the patients covered in this multicenter study, and clinical data and evaluations were collected from different physicians. Finally, vaccine type and onset of symptoms after first or second dose differs between the cases and are therefore difficult to compare. 

## 4. Materials and Methods

### 4.1. Patients and Clinical Investigations

In total, 15 patients (median age 38 (31–52) years; 9 men) from 11 different clinics were included in this this multicenter study. For all patients, the clinical suspicion of myocarditis after COVID-19 vaccination was expressed by the physician in charge due to a temporal relationship of the vaccination and the onset of myocarditis symptoms. All patients underwent key clinical investigations including laboratory testing, echocardiography and electrocardiogram, and some patients (*n* = 6) underwent additional cardiac magnetic resonance imaging (cMRI) [28]. Furthermore, EMBs were obtained from each patient and sent to the Institute for Cardiac Diagnostics and Therapy (IKDT, Berlin, Germany), for routine workup including histology, immunohistochemistry and molecular virology. The decision to perform EMB was made by the attending physician based on the setting of unexplained heart failure and sudden onset of severe left ventricular failure [29]. Coronary artery disease and other possible causes of myocardial dysfunction have been excluded by angiography prior to EMB in all patients. Left ventricular ejection fraction (LVEF) was determined by echocardiography with the Simpson method. All numerical data are presented with median and interquartile range.

### 4.2. Molecular Virology of EMBs

Immediately after withdrawal of EMBs, samples were transferred to RNAlater solution for nucleic acids stabilization (Thermo Fisher Scientific, Waltham, MA, USA). Total DNA was extracted by Gentra Puregene Kits (Qiagen, Hilden, Germany) according to manufacturer’s instructions [16]. Total RNA isolation was performed using TRIzol Reagent (Thermo Fisher Scientific, Waltham, MA, USA), followed by DNAse treatment (Promega, Walldorf, Germany) and cDNA synthesis using High-Capacity cDNA Reverse Transcription Kit (Thermo Fisher Scientific, Waltham, MA, USA) and random hexamer primers (Thermo Fisher Scientific, Waltham, MA, USA) [19]. Then, the detection of viral genomes including Parvovirus B19 (B19V), Enteroviruses (including Coxsackievirus B3 and Echoviruses), Adenoviruses, Epstein–Barr virus, human Herpesvirus 6 and Severe Acute Respiratory Syndrome Coronavirus 2 (SARS-CoV-2) was accomplished using nested polymerase chain reaction (nested-PCR) and quantitative reverse transcriptase (qRT)-PCR, as described before [19,30].

### 4.3. Histology, Immunohistochemistry and Digital Imaging Analysis

Histological examinations were carried out on formalin-fixed, paraffin-embedded specimens stained with hematoxylin and eosin (H & E) and trichrome stains according to standard procedures [31]. Active myocarditis was diagnosed according to the histomorphological Dallas criteria [32]. For qualification and quantification of inflammatory infiltrates, immunohistochemical staining was carried out on RNAlater-fixed, cryo-embedded EMBs, as described before [31]. Myocardial inflammation was diagnosed by the presence of ≥14 leucocytes with the presence of CD3^+^ T cells according to the European Society of Cardiology (ESC) position statement [33]. Furthermore, the number of CD11a^+^/LFA-1^+^ lymphocytes (threshold ≥ 14 cells/mm^2^), CD11b^+^/MAC-1^+^ macrophages (threshold ≥ 40 cells/mm^2^), CD45R0^+^ T-memory cells (threshold ≥ 50 cells/mm^2^), perforin^+^ cytotoxic cells (threshold ≥ 2.9 cells/mm^2^), and HLA-DR^+^-presenting cells (threshold ≥ 4.6 area%) was determined. In addition, ratio of CD4^+^-to-CD8^+^ T cells was analyzed. Tissue and endothelial activation were measured by the expression of the intercellular adhesion molecule 1 (ICAM-1, threshold ≥ 2.8 area%) and vascular cell adhesion molecule 1 (VCAM-1, threshold ≥ 0.08 area%). All immunohistochemical markers were quantified using digital imaging analysis, as described previously [23]. Detection of SARS-CoV-2 spike protein was performed on paraffin-embedded EMBs using an appropriate antibody (GeneTex, 1A9, GTX632604; 1:100) [34]. Autopsy cardiac tissue from a SARS-CoV-2 qRT-PCR-confirmed patient was used as positive control.

## 5. Conclusions

The present study summarizes EMB-based diagnostics of 15 patients with clinical suspicion of myocarditis following vaccination against SARS-CoV-2. It identifies 14 of 15 patients with myocardial inflammation, ranging from inflammatory cardiomyopathy to active myocarditis and severe giant cell myocarditis. Although a causal relationship between vaccination and the occurrence of myocardial inflammation cannot be established based on the findings, the cardiac detection of spike protein, the CD4^+^ T-cell-dominated inflammation and the close temporal relationship argue for a vaccine-triggered autoimmune reaction.

## Figures and Tables

**Figure 1 ijms-23-06940-f001:**
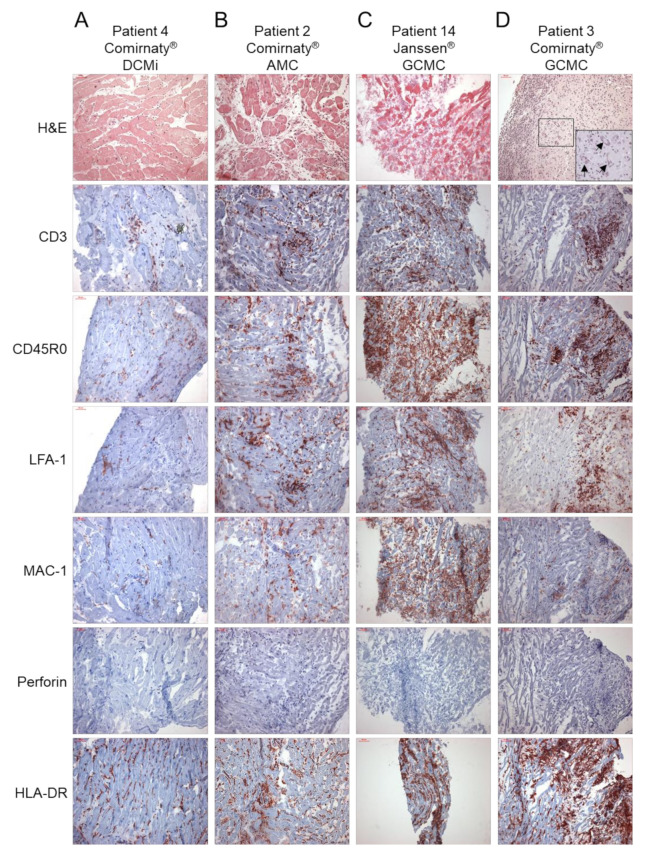
Representative images of haematoxylin and eosin (H & E) staining, and immunohistochemical stainings for the assessment of inflammation in endomyocardial biopsies from patients with the suspicion of myocarditis after COVID-19 vaccination. Immunohistochemical detection of CD3^+^ T cells, CD45R0^+^ T-memory cells, LFA-1^+^ lymphocytes, MAC-1^+^ macrophages, perforin^+^ cytotoxic cells and HLA-DR^+^ activated T cells in patients diagnosed for (**A**) inflammatory cardiomyopathy (DCMi, patient 4, Comirnaty^®^ vaccine), (**B**) acute myocarditis (AMC, patient 2, Comirnaty^®^ vaccine) and (**C**,**D**) giant cell myocarditis (GCMC, patient 14, Janssen^®^ vaccine; patient 3, Cormirnaty^®^ vaccine; giant cells are marked by arrows in H & E staining). Immunohistochemical staining was quantified by digital image analysis and is depicted for each patient in Table 2. Magnification 200×. Scale bars 50 μm.

**Figure 2 ijms-23-06940-f002:**
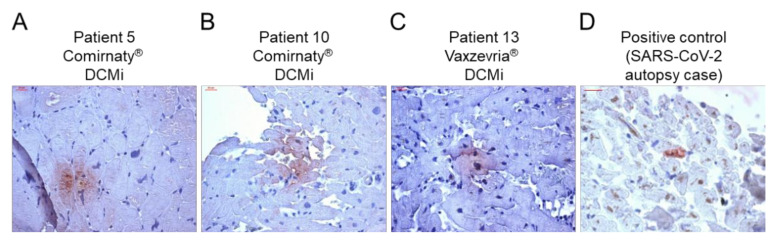
Evidence of SARS-CoV-2 spike protein in cardiac tissue after COVID-19 vaccination. (**A**–**C**) Representative immunohistochemical stainings of SARS-CoV-2 spike protein in EMBs from patients diagnosed with DCMi after receiving Comirnaty^®^ (panel A and B, patients 5 and 10) or Vaxzevria^®^ (panel C, patient 13). (**D**) SARS-CoV-2-positive cardiac tissue served as positive control. Magnification 400×. Scale bars 20 μm.

**Figure 3 ijms-23-06940-f003:**
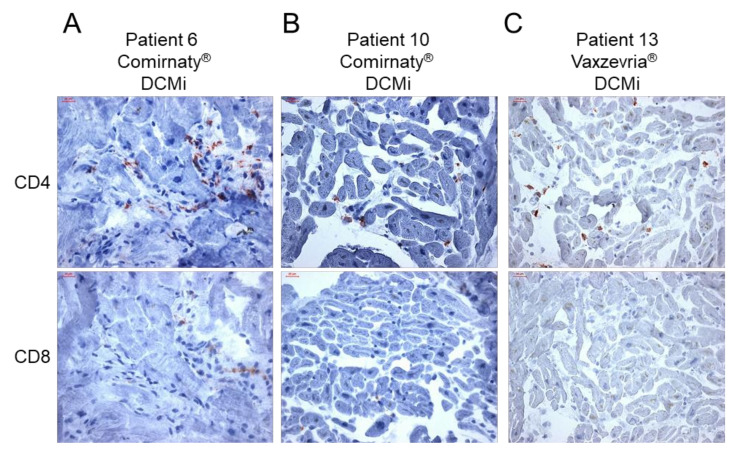
Inflammatory cardiomyopathy in response to COVID-19 vaccination is dominated by CD4^+^ T cells. (**A**–**C**) Representative immunohistochemical stainings of CD4^+^ and CD8^+^ T cells in endomyocardial biopsies from patients diagnosed for inflammatory cardiomyopathy (DCMi) after receiving Comirnaty^®^ (panel A and B, patients 6 and 10) or Vaxzevria^®^ (panel C, patient 13) vaccines, respectively. Immunohistochemical staining was quantified by digital image analysis and CD4-to-CD8 ratio is depicted for each patient in Table 2. Magnification 400×. Scale bars 20 μm.

**Table 1 ijms-23-06940-t001:** Patients’ characteristics, data on vaccination, clinical findings and EMB based diagnosis.

Pat. No.	Sex	Age (y)	LVEF (%)	Vaccine	Manufacturer	Dose	Onset of Symptoms (Days)	Clinical Picture	Troponin (pg/mL) Normal <15	BNP (pg/mL) Normal <125	CK (U/L) Normal <171	CRP (mg/dL) Normal <0.5	Suspected Diagnosis	Diagnosis
1	m	23	45	Comirnaty	Pfizer-BioNTech	2nd	20	discomfort during exercise, reduced LVEF, dilated LV	<15			<0.5	AMC; myocarditis after vaccination	AMC
2	f	31	20	Comirnaty	Pfizer-BioNTech	2nd	0	cardiac arrest during sports 6 h after the 2nd vaccination, resuscitation, dyspnea, hopotonia, reduced LVEF, dilated LV, NYHA III, intrapulmonary infiltrates, pericardial effusion, no signs of active myocarditis in cMRI	1325	2183	784	7.3	AMC; myocarditis after vaccination	AMC
3	m	32	43	Comirnaty	Pfizer-BioNTech	1st	1–3	dyspnea, reduced exercise capacity, reduced LVEF, NYHA II	576	2483		8.9	AMC; myocarditis after vaccination	GCMC
4	m	52	45	Comirnaty	Pfizer-BioNTech	2nd	3	inpatient admission after resuscitation for ventricular fibrillation, reduced LVEF, LV latero-apical akinesia with wall thinning, no signs of active myocarditis in cMRI	436			2.8	ARVC; myocarditis after vaccination	DCMi
5	m	18	12	Comirnaty	Pfizer-BioNTech	2nd	21	cardiac decompensation, reduced LVEF, NYHA II-III, no signs of active myocarditis in cMRI	38.1	8430	181	0.7	DCM; myocarditis after vaccination	DCMi
6	m	59	38	Comirnaty	Pfizer-BioNTech	2nd	56	dyspnea, reduced LVEF	58.3	669			myocarditis after vaccination	DCMi
7	m	24	30	Comirnaty	Pfizer-BioNTech	2nd	2	dyspnea, angina pectoris, reduced LVEF	710		585	9.3	AMC; myocarditis after vaccination	DCMi
8	m	39	5	Comirnaty	Pfizer-BioNTech	2nd	4	dyspnea, cardiac decompensation, reduced LVEF, signs of active myocarditis in cMRI		935	68	1.47	AMC; myocarditis after vaccination	DCMi
9	m	34	10	Comirnaty	Pfizer-BioNTech	1st	14	dyspnea, reduced exercise capacity, reduced LVEF, dilated LA, supraventricular tachycardia up to 140/min, atrial fibrillation, myocardial edema and signs of active myocarditis in cMRI	<15				AMC; myocarditis after vaccination	DCMi
10	f	38	40	Comirnaty	Pfizer-BioNTech	2nd	14	reduced LVEF, NYHA I	<15	84	35	2.8	AMC; myocarditis after vaccination	DCMi
11	f	52	15	Comirnaty	Pfizer-BioNTech	2nd	1	dyspnea on exertion, reduced LVEF, mitral valve insufficiency, hypertension, no signs of active myocarditis in cMRI		1592	83	7.1	AMC; myocarditis after vaccination	DCMi
12	f	59	37	Vaxzevria	AstraZenica	2nd	14	dyspnea, reduced LVEF	<15				myocarditis after vaccination	DCM
13	f	68	30	Vaxzevria	AstraZenica	1st	1	reduced LVEF	582	1094			AMC; DCMi; myocarditis after vaccination	DCMi
14	f	31	35	Janssen	Johnson & Johnsen	1st	28	fulminant cardiogenic shock, reduced LVEF	48	334	1557	35	AMC; GCMC; Sarkoidosis; myocarditis after vaccination	GCMC
15	m	45	10	Janssen	Johnson & Johnsen	1st	14	dyspnea, reduced LVEF, dilated LV, atrial fibrillation	23.7	1670	90	1.4	AMC; myocarditis after vaccination	DCMi

AMC, active myocarditis according to the Dallas criteria; BNP, brain natriuretic peptide; CK, creatinine kinase; cMRI, cardiac magnetic resonance imaging; CRP, C-reactive protein; DCM, dilated cardiomyopathy; DCMi, inflammatory cardiomyopathy; f, female; GCMC, giant cell myocarditis; LA, left atrium; LV, left ventricle; LVEF, left ventricular ejection fraction; m, male; NYHA, New York heart association; RA, right atrium; RV, right ventricle.

**Table 2 ijms-23-06940-t002:** Biopsy findings. Virological analysis includes detection of genomes from SARS-CoV-2, Parvovirus B19 (B19V), Enterovirus, Adenovirus, human Herpesvirus 6 and Epstein–Barr virus. Immunohistochemical analysis includes detection of CD3^+^, CD4^+^ and CD8^+^ T cells, CD45R0^+^ T-memory cells, LFA-1^+^ lymphocytes, MAC-1^+^ macrophages, perforin^+^ cytotoxic cells, HLA-DR^+^-presenting cells, ICAM-1^+^ and VCAM-1^+^ endothelial cell-adhesion molecules, and SARS-CoV-2 spike protein. All inflammation markers were quantified by digital image analysis and are depicted as cells/mm^2^ or area %, respectively. Spike protein was evaluated as not expressed (− or sparsely expressed (+).

Pat. No.	Vaccine	Diagn.	Ventr.	Virology	CD3 (Cells/mm^2^) Normal <14	CD45R0 (Cells/mm^2^) Normal <60	LFA-1 (Cells/mm^2^) Normal <14	MAC-1 (Cells/mm^2^) Normal <40	Perforin (Cells/mm^2^) Normal <3.5	HLA-DR (Area%) Normal <4.6	ICAM-1 (Area%) Norma <2.8	VCAM-1 (Area%) Normal <0.08	CD4 to CD8 Ratio	SARS-CoV-2 Spike Protein
1	Comirnaty	AMC	RV	B19V	11.3	33.0	17.2	34.7	1.1	4.7	2.2	0.02	0.7	-
2	Comirnaty	AMC	LV	-	500	749	501	277	0.0	6.2	4.8	0.61	0.2	+
3	Comirnaty	GCMC	LV	-	274	682	439	80.6	0.0	12.7	nd	nd	0.7	+
4	Comirnaty	DCMi	LV	B19V	52.1	132.6	45.4	86.5	0.0	5.6	0.3	0.01	2.0	-
5	Comirnaty	DCMi	LV	B19V	4.8	51.6	18.3	38.3	0.0	3.4	0.7	0.01	1.0	+
6	Comirnaty	DCMi	LV	B19V	17.9	67.3	6.7	38.1	0.0	3.8	nd	nd	7.4	-
7	Comirnaty	DCMi	LV	-	21.4	110	95.5	53.3	0.0	5.0	2.1	0.04	1.0	+
8	Comirnaty	DCMi	LV	B19V	18.3	153	24.8	108	0.0	6.6	0.7	0.02	2.8	-
9	Comirnaty	DCMi	RV	B19V	48.7	62.2	72.8	84.3	0.0	8.0	2.2	0.02	0.6	-
10	Comirnaty	DCMi	LV	B19V	7.1	79.8	21.9	54.4	0.0	4.4	nd	nd	11.2	+
11	Comirnaty	DCMi	RV	B19V	10.5	76.7	28.4	61.3	2.5	8.2	nd	nd	1.0	+
12	Vaxzevria	DCM	RV	B19V	2.9	31.4	8.8	28.6	0.0	2.9	0.6	0.00	1.0	+
13	Vaxzevria	DCMi	LV	-	13.4	114	13.4	81.3	1.8	12.3	nd	nd	12.4	+
14	Janssen	GCMC	LV	-	800	987	713	360	0.0	23.1	6.4	0.37	0.6	nd
15	Janssen	DCMi	RV	B19V	52.9	108	54.0	100	0.0	8.0	3.7	0.06	1.0	+

AMC, active myocarditis according to the Dallas criteria; B19V, latent Parvovirus B19 infection; DCM, dilated cardiomyopathy; DCMi, inflammatory cardiomyopathy; GCMC, giant cell myocarditis; LV, left ventricle; RV, right ventricle; nd, not determined.

## Data Availability

All authors confirm that all related data supporting the findings of this study are given in the article.

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
