# Peer review of "Intramyocardial Inflammation after COVID-19 Vaccination: An Endomyocardial Biopsy-Proven Case Series"

_ijms, 2022, doi:10.3390/ijms23136940_

Round 1

Reviewer 1 Report

The present study summarizes EMB-based diagnostics of 15 patients with the clinical suspicion of myocarditis following vaccination against SARS-CoV-2. It identifies 14 of 15 patients with myocardial inflammation. The authors conclude that these findings might be due to a vaccine-triggered reaction. 

The study is well written and the results are interesting. I do have some minor comments that should be addressed:

  • I agree that EMB is the gold standard for the diagnosis of inflammatory cardiomyopathies. However, due to its invasive nature, costs, and risks (low but still present), it has been seldom performed in hemodynamically stable patients with suspected acute myocarditis (when non-invasive imaging techniques are usually preferred). The authors should clarify the reason why EMB was performed in all patients (in particular those other 10 without severe complications).
  • It would be interesting to show troponins and CRP values for all patients. Why are there so many missing Troponin and CRP data in Table 1? It is unlikely that these biomarkers were not measured in case of suspected myocarditis.
  • in the same way, it would be helpful to support the conclusions to show a panel with the results of the myocarditis antibody panel. Was a real-time polymerase chain reaction for SARS-CoV5 2 performed?
  • In some recent publications, myocarditis following mRNA COVID-19 vaccination has been showed using somatostatin receptor PET/CT. I suggest to highlight and discuss the relevance of non-invasive imaging techniques in the diagnostic work-up of this new subtype of myocarditis.

Author Response

First of all, we would like to thank reviewer 1 for his/her helpful and constructive suggestions which we tried to follow. In response to his/her comments, we have implemented the suggested changes as listed below:

I agree that EMB is the gold standard for the diagnosis of inflammatory cardiomyopathies. However, due to its invasive nature, costs, and risks (low but still present), it has been seldom performed in hemodynamically stable patients with suspected acute myocarditis (when non-invasive imaging techniques are usually preferred). The authors should clarify the reason why EMB was performed in all patients (in particular those other 10 without severe complications).

We thank reviewer 1 for this question and included a statement about the rationale to perform EMB diagnostics for each patient in the Materials and Methods part.

It would be interesting to show troponins and CRP values for all patients. Why are there so many missing Troponin and CRP data in Table 1? It is unlikely that these biomarkers were not measured in case of suspected myocarditis.

Due to the multicenter character of this study, we tried to compile all clinical data, including Troponin and CRP levels, in the manuscript. Unfortunately, some data on Troponin or CRP were not available and are therefore not included in table 1. However, data from patients # 4, 5, 8, 10, 12 and 14 were requested and are now included in table 1.

In the same way, it would be helpful to support the conclusions to show a panel with the results of the myocarditis antibody panel. Was a real-time polymerase chain reaction for SARS-CoV-2 performed?

The results of the immunohistochemical analysis of EMB specimen are shown in table 2. Here, all applied antibodies for the verification of the myocardial inflammation are depicted with appropriate quantitative results (as cells/mm² or area%/mm², respectively), and representative staining’s of all antibodies are presented in Figures 1 and 2. Indeed, we have performed a quantitative real-time polymerase chain reaction for SARS-CoV-2 with EMBs from all included patients. We state this in the Results and Materials and Methods sections.

In some recent publications, myocarditis following mRNA COVID-19 vaccination has been showed using somatostatin receptor PET/CT. I suggest to highlight and discuss the relevance of non-invasive imaging techniques in the diagnostic work-up of this new subtype of myocarditis. 

We thank the reviewer for his/her valuable comment. We agree, that advanced imaging techniques play a crucial role in the diagnostics of myocarditis. Most of the published cases of myocarditis after COVID-19 vaccination were indeed diagnosed by cardiac imaging, however, EMB-proven cases are limited. Here, we concentrate on the EMB-based histopathological analysis and compare this to other available clinical parameters (incl. cMRI). To improve the discussion, we added a chapter about non-invasive imaging techniques.

Reviewer 2 Report

In this manuscript, Baumeier C. et al. present a histopathological analysis of endomyocardial biopsy from 15 patients with reduced ejection fraction and the clinical suspicion of myocarditis following SARS Cov-2 vaccination with Comirnaty (Pfizer-BioNTech), Vaxzevria (AstraZenica) or Janssen (Johnson & Johnson). After excluding possible infectious causes, the Immunohistochemical analyses reveal myocardial inflammation in 14 of 15 patients with a histopathological diagnosis of active myocarditis according to the Dallas criteria in 2 patients, severe giant cell myocarditis in 2 patients and inflammatory cardiomyopathy in 10 patients. The authors concluded that although, a definitive causal relationship between SARS Cov-2 vaccination and the occurrence of myocardial inflammation cannot be demonstrated, data suggests a temporal correlation.

I read with great interest this paper about the histopathological characteristics of suspected myocarditis induced by SARS Cov-2 vaccines. However, I think that some aspects of this study should be clarified in depth before thinking about their publication in an open access journal with a high impact factor as IJMS.

After mass vaccination with mRNA vaccines against SARS Cov-2, myocarditis in male teenagers emerged as a possible rare side effect, determining in the general population hesitation or refusal of vaccination, and fueling controversy against the scientific community.

For this reason I think that any study on this topic must be methodologically very accurate in order to avoid possible misunderstandings and exploitation.

My main concerns are the following:

Major point

  • The design of the study needs to be clarified and stated in the manuscript title. In my opinion it is simply a case series with a very accurate histopathological characterization.

  • It is unclear how patients were selected. They are consecutive patients enrolled at the same center? Are they patients from different centers?

  • How was the clinical suspicion of vaccine myocarditis postulated? This should be clearly expressed in the methods by dedicating a paragraph to it. The authors talk about a temporal correlation, but it is not clear what it is. The onset of symptoms varies from 0 to 56 days;Please, justify the time range with appropriate citations.

  • Table 1 shows for each patient the signs and symptoms associated with possible myocarditis. The data reported are very heterogeneous. (ie: Table 1, patient n 4: what do you mean for “..inhomogeneous muscular septal structure”? Table 1, patient n 11: what do you mean for: “ .. diastolic insufficiency”? Table 1, patient n 12 :” dyspnea, reduced respiratory rate”. Was the patient in shock with cerebral hypoperfusion? Have other causes of central dyspnea been ruled out? Table 15, patients n 15: by tachyarrhythmia absoluta do you mean atrial fibrillation?). In the methods should accurately list the clinical and instrumental signs and symptoms associated with myocarditis. Justifying how the clinical suspicion was postulated. Please, also add a reference regarding clinical signs and symptoms associated with myocarditis.

  • I do not understand after how many doses of vaccine the myocarditis occurred in each patient. This information should be reported.

  • The results show the following data: “Mean left ventricular ejection fraction (LVEF) was 27.7 ± 14.3%”. How was it calculated? by ultrasound? with which method (i.e. Simpson’s method)? Is there any other data regarding cardiac structure? If the evaluation was done by ultrasound, were all the examinations performed by the same operator?

  • Did the patients have previous cardiovascular disease?

  • Please restate the conclusions according to the purpose of the study. If you cannot verifiably describe a causal relation of vaccination and the occurrence of inflammation in EMB you cannot conclude that:” the benefit of vaccination outweighs its possible adverse events”. This is a descriptive study of histopathological aspects, and conclusions should be limited to these aspects only.

Minor point:

  • Probably due to a layout problem, the discussion seems to be divided into two parts starting with line 162. please unify the text

  • I think a study limitations section should be added to the manuscript

Author Response

Thank you for the constructive comments/suggestions and for providing confidence to our work. We tried to follow your suggestions as follows.

The design of the study needs to be clarified and stated in the manuscript title. In my opinion it is simply a case series with a very accurate histopathological characterization.

We totally agree and changed the title to: “Intramyocardial inflammation after COVID-19 vaccination: An endomyocardial biopsy-proven case series”

It is unclear how patients were selected. They are consecutive patients enrolled at the same center? Are they patients from different centers?

The present work is a multicenter study, which was combined in EMB diagnostics at the Institute for Cardiac Diagnostics and Therapy (IKDT). EMBs from patients with the suspicion of vaccine-mediated myocarditis were taken at different clinics and sent to IKDT for differential diagnostics. Thus, patients were selected according to the clinical suspicion and the availability of EMBs for diagnostic purpose. We included a statement about patient inclusion criteria in the Materials and Methods part.

How was the clinical suspicion of vaccine myocarditis postulated? This should be clearly expressed in the methods by dedicating a paragraph to it. The authors talk about a temporal correlation, but it is not clear what it is. The onset of symptoms varies from 0 to 56 days; Please, justify the time range with appropriate citations.

The clinical suspicion of vaccine myocarditis was made by the physician in charge based on key clinical investigations including laboratory testing, echocardiography, electrocardiogram, and in some cases additional cardiovascular magnetic resonance imaging. In addition, a temporal relationship between COVID-19 vaccination and the onset of symptoms were given in all cases. We state this in chapter 4.1. “Patients and clinical investigations of myocarditis”. Furthermore, we discuss the range of symptoms onset with appropriate citations in the Discussion part.

Table 1 shows for each patient the signs and symptoms associated with possible myocarditis. The data reported are very heterogeneous. (ie: Table 1, patient n 4: what do you mean for “..inhomogeneous muscular septal structure”? Table 1, patient n 11: what do you mean for: “ .. diastolic insufficiency”? Table 1, patient n 12 :” dyspnea, reduced respiratory rate”. Was the patient in shock with cerebral hypoperfusion? Have other causes of central dyspnea been ruled out? Table 15, patients n 15: by tachyarrhythmia absoluta do you mean atrial fibrillation?). In the methods should accurately list the clinical and instrumental signs and symptoms associated with myocarditis. Justifying how the clinical suspicion was postulated. Please, also add a reference regarding clinical signs and symptoms associated with myocarditis.

We agree with the reviewer, our data are heterogeneous because the patients showed quite different clinical manifestations and the data collection came from different clinics. We therefore modified the table 1. In addition, we have revised the Materials and Methods section regarding the clinical diagnostics of myocarditis and included an appropriate reference in chapter 4.1. “Patients and clinical investigations of myocarditis”. Moreover, we added a study limitation part to include the passage that the clinical data were heterogeneous and collected by different physicians. 

I do not understand after how many doses of vaccine the myocarditis occurred in each patient. This information should be reported.

We regret to forget this important information. We included the data on number of doses in table 1.

The results show the following data: “Mean left ventricular ejection fraction (LVEF) was 27.7 ± 14.3%”. How was it calculated? by ultrasound? with which method (i.e. Simpson’s method)? Is there any other data regarding cardiac structure? If the evaluation was done by ultrasound, were all the examinations performed by the same operator?

Left ventricular ejection fraction (LVEF) was determined in each case by ultrasound using Simpson's method. Since this study is multicenter and the patients come from different hospitals, the echocardiographic findings have been collected by different investigators. We have highlighted this difference in clinical data collectionin the Study Limitation section in our manuscript.

Did the patients have previous cardiovascular disease?

There was no previous cardiovascular disease reported in any of the patients.

Please restate the conclusions according to the purpose of the study. If you cannot verifiably describe a causal relation of vaccination and the occurrence of inflammation in EMB you cannot conclude that:” the benefit of vaccination outweighs its possible adverse events”. This is a descriptive study of histopathological aspects, and conclusions should be limited to these aspects only.

We agree, that this statement is not supported by own data and therefore moved this issue to the Discussion part.

Probably due to a layout problem, the discussion seems to be divided into two parts starting with line 162. please unify the text.

We very much appreciate this comment of the reviewer and confirm, that it was an initial problem of formatting. We have fixed this problem in the revised manuscript.

I think a study limitations section should be added to the manuscript.

We are absolutely with this comment and included a section about Study Limitations at the end of the Discussion.

Reviewer 3 Report

- Add to the title that this is a “case series”
- As the limited number of patients reported, I strongly suggest to express aggregate data as median and interquartile range (IQR)
- The section Results should summarize (and not repeat) what is reported in Tables. Please highlight only the most relevant findings or report aggregate observations (as median and IQR).
- L219-220 must be moved in Results section
- L260-263 this paragraph is not related with the main topic, and as this is not an epidemiological study, it is not supported by findings.

Author Response

First of all, we would like to thank reviewer 3 for his/her helpful and constructive suggestions which we tried to follow. In response to his/her comments, we have implemented the suggested changes as listed below:

Add to the title that this is a “case series”.

We agree with reviewer 3 and changed the title to: “Intramyocardial inflammation after COVID-19 vaccination: An endomyocardial biopsy-proven case series”

As the limited number of patients reported, I strongly suggest to express aggregate data as median and interquartile range (IQR).

We totally agree with the reviewer that the number of patients is too little to express aggregate data as mean ± SD. Consequently, data on age and LVEF were changed and are now expressed as median ± IQR.

The section Results should summarize (and not repeat) what is reported in Tables. Please highlight only the most relevant findings or report aggregate observations (as median and IQR).

We thank the reviewer for his/her valuable comment. Accordingly, the Results part was shortened and only most relevant findings are reported as median ± IQR.

L219-220 must be moved in Results section.

We would like to thank reviewer 3 for this advice. The sentence referring to table 1 had been moved to the Results section.

L260-263 this paragraph is not related with the main topic, and as this is not an epidemiological study, it is not supported by findings.

We agree that this statement is not supported by own data and therefore moved this issue to the Discussion.

Round 2

Reviewer 2 Report

I appreciated the changes made to the manuscript by the authors however there are still many inaccuracies both methodological and conceptual.

In particular, my major concerns are:

  • The conclusion of the abstract still carries concepts that the authors cannot state with the available data (lines:46-49):” Although, a definitive causal relationship between COVID-19 vaccination and the occurrence of myocardial inflammation cannot be demonstrated in this study, data suggests a temporal correlation. Nevertheless, further investigations are needed to clarify the underlying pathophysiological mechanisms”. What is the purpose of the study? To present a comprehensive histopathological analysis of EMBs from patients with the clinical suspicion of myocarditis following vaccination or to demonstrate a temporal correlation between vaccination and myocarditis? If the purpose of the study is to describe the histopathological aspect of EMBs, the conclusions should be limited solely and exclusively to the histopathological aspects of the biopsies

  • Given the low sample size, reviewer 3 correctly requested that the values of the linear variables be expressed as median with interquartile range. However, the authors reported the median but not the IQR. IQR cannot be expressed as ± followed by a single number. IQR is a range between the first and the fourth quartile. This implies that the numbers to be reported are 2. This error shows a superficial knowledge of the basic principles of statistics.

  • Table 1: the clinical information regarding clinical features associated with suspected myocarditis is too confusing. There are anamnestic data, clinical signs, alterations found on instrumental investigations, and brief reports of clinical course (as well as inaccuracies and repetition)

  • Although I completely agree with what is stated in the lines 232-235, moving these statements from the conclusions to the discussion does not solve the problem posed in the previous review. In this work, the authors have not demonstrated anything that would confirm or disprove these issues

  • Lines 305-307: see point 1

Although I continue to find the histopathological data shown in this paper very interesting, I think that the way in which the data are expressed and discussed still needs a deep reorganization. For these reasons, I do not believe the work has sufficient quality to be published in this journal.

Author Response

We would like to thank reviewer 2 for his/her additional revision of our manuscript including his/her helpful suggestions, which we tried to follow. In response to his/her comments, we have implemented the suggested changes as listed below:

The conclusion of the abstract still carries concepts that the authors cannot state with the available data (lines:46-49):” Although, a definitive causal relationship between COVID-19 vaccination and the occurrence of myocardial inflammation cannot be demonstrated in this study, data suggests a temporal correlation. Nevertheless, further investigations are needed to clarify the underlying pathophysiological mechanisms”. What is the purpose of the study? To present a comprehensive histopathological analysis of EMBs from patients with the clinical suspicion of myocarditis following vaccination or to demonstrate a temporal correlation between vaccination and myocarditis? If the purpose of the study is to describe the histopathological aspect of EMBs, the conclusions should be limited solely and exclusively to the histopathological aspects of the biopsies.

The purpose of this study was to provide a comprehensive histopathological analysis of EMBs from patients with the suspicion of myocarditis following COVID-19 vaccinations. This analysis revealed myocardial inflammation in 14 of 15 cases with a temporal connection to the vaccination ranging from 0-56 days (median 14 days). Therefore, we disagree that the conclusion “Although, a definitive causal relationship between COVID-19 vaccination and the occurrence of myocardial inflammation cannot be demonstrated in this study, data suggests a temporal correlation. Nevertheless, further investigations are needed to clarify the underlying pathophysiological mechanisms” is not supported by the presented data. However, we revised the conclusion regarding the focus on novel histopathological data on spike protein expression and autoimmunological aspects, and tried to deflect the attention from the temporal connection.

Given the low sample size, reviewer 3 correctly requested that the values of the linear variables be expressed as median with interquartile range. However, the authors reported the median but not the IQR. IQR cannot be expressed as ± followed by a single number. IQR is a range between the first and the fourth quartile. This implies that the numbers to be reported are 2. This error shows a superficial knowledge of the basic principles of statistics.

We apologize for this mistake and now present all numerical data with median and interquartile range.

Table 1: the clinical information regarding clinical features associated with suspected myocarditis is too confusing. There are anamnestic data, clinical signs, alterations found on instrumental investigations, and brief reports of clinical course (as well as inaccuracies and repetition)

We express regret, that the clinical data shown in table 1 are confusing and partly redundant. We reviewed the table and present solely data of clinical importance in the revised manuscript.

Although I completely agree with what is stated in the lines 232-235, moving these statements from the conclusions to the discussion does not solve the problem posed in the previous review. In this work, the authors have not demonstrated anything that would confirm or disprove these issues

We agree with the reviewers’ comment and removed this sentence from the manuscript.

Lines 305-307: see point 1

See comment on point 1.